# Whole Conversion of Soybean Molasses into Isomaltulose and Ethanol by Combining Enzymatic Hydrolysis and Successive Selective Fermentations

**DOI:** 10.3390/biom9080353

**Published:** 2019-08-09

**Authors:** Zhi-Peng Wang, Lin-Lin Zhang, Song Liu, Xiao-Yan Liu, Xin-Jun Yu

**Affiliations:** 1Marine Science and Engineering College, Qingdao Agricultural University, Qingdao 266109, China; 2College of Chemistry & Environmental Engineering, Shandong University of Science & Technology, Qingdao 266510, China; 3Development & Reform Bureau, West Coast New Area, Qingdao 266000, China; 4Jiangsu Key Laboratory for Biomass-based Energy and Enzyme Technology, Huaiyin Normal University, Huaian 223300, China; 5Key Laboratory of Bioorganic Synthesis of Zhejiang Province, College of Biotechnology and Bioengineering, Zhejiang University of Technology, Hangzhou 310014, China

**Keywords:** isomaltulose, soy molasses, raffinose-family oligosaccharides, α-galactosidase, selective fermentation

## Abstract

Isomaltulose is mainly produced from sucrose by microbial fermentation, when the utilization of sucrose contributes a high production cost. To achieve a low-cost isomaltulose production, soy molasses was introduced as an alternative substrate. Firstly, α-galactosidase gene from *Rhizomucor miehei* was expressed in *Yarrowia lipolytica,* which then showed a galactosidase activity of 121.6 U/mL. Under the effects of the recombinant α-galactosidase, most of the raffinose-family oligosaccharides in soy molasses were hydrolyzed into sucrose. Then the soy molasses hydrolysate with high sucrose content (22.04%, *w*/*w*) was supplemented into the medium, with an isomaltulose production of 209.4 g/L, and the yield of 0.95 g/g. Finally, by virtue of the bioremoval process using *Pichia stipitis*, sugar byproducts in broth were transformed into ethanol at the end of fermentation, thus resulting in high isomaltulose purity (97.8%). The bioprocess employed in this study provides a novel strategy for low-cost and efficient isomaltulose production from soybean molasses.

## 1. Introduction

Isomaltulose is a kind of structural isomer of sucrose and natural reducing disaccharide. Although isomaltulose shares similar physical and organoleptic properties with sucrose, it has been approved as a safer sucrose substitute with advantages including slower release and absorption, higher stability, lower plasma glucose and insulin concentrations [1,2,3]. In addition, due to low hygroscopicity, acid stability and reducing properties, isomaltulose can also be used to produce food and industrial products, like instant powder products, sports beverages, and isomaltitol [4,5,6].

Since chemical synthesis of isomaltulose is accompanied with some byproducts that are difficult to remove, bacterial fermentation became the primary technology adopted in industrial isomaltulose synthesis [7,8]. The bacteria secreted a type of enzyme called sucrose isomerase (SIase), isomerizing sucrose into isomaltulose in the medium [9]. However, bacterial fermentation faced two weaknesses: strain safety and low production efficiency. Expressing SIase genes in food-grade hosts previously received more attention in order to solve the weaknesses [9]. Some non-invasive and non-pathogenic microbes, like *Lactococcus lactis*, *Bacillus subtilis*, *Saccharomyces cerevisiae*, and *Yarrowia lipolytica,* have been employed as expression hosts. Using the engineered strains, increased SIase activity and maintained security were achieved [8,10,11,12,13]. In particular, using the engineered *Y. lipolytica* strain, in which a bacteria-derived SIase gene was expressed extracellularly, an isomaltulose production of 572.1 g/L from sucrose was achieved and the yield reached 0.96 (g/g), suggesting a huge potential for industrial isomaltulose production [13].

Another limiting factor for isomaltulose production is the relatively expensive price of the sucrose substrate. The cost of sucrose takes up more than 50% of the total cost. Therefore, introducing inexpensive feedstocks instead of pure sucrose as alternative substrate was economically feasible to lower the cost [8,13,14]. Cane molasses, as a well-studied waste of sugar industry with a sucrose content of more than 30%, have been used for synthesis of isomaltulose by *Bacillus subtilis*.

Soy molasses is a low-value byproduct obtained during the extraction of soybean proteins and was generated largely with the increasing demand of high-quality soy protein [15]. Sugar content of soy molasses is usually more than 30% (*w*/*w*), containing mainly stachyose, sucrose, raffinose, glucose, xylose, galactose, and fructose. Due to the high carbohydrate content and the relatively low cost, soy molasses have been of great interest among researchers and biorefinery industrialists to produce value-added chemicals. However, the current soy molasses-derived fermentations have limited efficiency, as most have focused on the utilization of the single constituents. Due to the high content of indigestible raffinose-family oligosaccharides (RFOs), referring to stachyose and raffinose, it was difficult for soy molasses to be completely degraded. The RFOs of soy molasses could not be converted into a usable product, and thus was discharged as waste from the process [15,16,17,18,19]. Soy molasses was considered to be of much lower value than cane molasses [16,19].

To increase the utilization efficiency of soy molasses, fermentation and complex enzymatic methods have been adopted to remove the RFOs and convert them into digestible monosaccharides. Notably, in particular, the single enzyme α-galactosidase can hydrolyze RFOs into sucrose and galactose [20]. α-galactosidase catalyzes the release of α-linked galactose residues from different substrates. Therefore, α-galactosidase has been widely used to remove the indigestible RFOs in the RFOs-rich food [20,21,22]. In addition to the native content of sucrose, the raised content of soluble sucrose makes soy molasses another low-cost alternative substrate for isomaltulose biosynthesis.

In this study, we sought to achieve efficient isomaltulose production from soy molasses. A heterologously expressed α-galactosidase was introduced to digest soy molasses and increase sucrose content. Then transformed and native sucrose in soy molasses was served as substrate to produce isomaltulose using an engineered *Y. lipolytica* strain. Subsequently, an ethanol producing bioremoval process was employed to remove sugar byproducts and improve isomaltulose purity. These processes presented a new potential choice for low-cost isomaltulose synthesis in industrial production.

## 2. Materials and Methods

### 2.1. Strains, Plasmids and Media

Uracil mutant strain *Y. lipolytica ura^-^* was employed as a host to express α-galactosidase. Recombinant strain S47 used for isomaltulose production has been described previously [13]. Four ethanol-producing strains identified as *Pichia stipitis, Saccharomyces cerevisiae, Hanseniaspora uvarum*, and *Kluyveromyces marxianu,* were preserved in the lab. The α-galactosidase gene used in this study was from *Rhizomucor miehei* (accession number: JF340459) and named as RMgase. RMgase gene manipulation was conducted in strain *E. coli* DH5α which was cultured in the Luria–Bertani (LB) medium. LB medium with the final concentration of 30.0 μg/mL kanamycin or 100.0 μg/mL ampicillin was prepared if necessary. YPD medium (20.0 g/L glucose, 20.0 g/L peptone, 10.0 g/L yeast extract) was used for culturing yeast strains. YNB plate (1.7 g/L yeast nitrogen base without amino acids, 10.0 g/L glucose, 5.0 g/L (NH4)_2_SO_4_, 25.0 g/L agar) was used for yeast transformant selection [23]. GPPB medium was used for SIase production and contained 30.0 g/L glucose, 1.0 g/L (NH4)_2_SO_4_, 6.0 g/L yeast extract, 2.0 g/L KH_2_PO_4_, 3.0 g/L K_2_HPO_4_, and 0.1 g/L MgSO_4_·7H_2_O with a pH of 6.0 [13].

### 2.2. Expression of RMgase Gene in Y. lipolytica

The construction of the expression vector pINA1312 containing RMgase gene was executed according to the literature description [13]. RMgase gene bearing the XPR2 (Extracellular alkaline protease gene from *Y. lipolytica*) signal peptide at its 5′ end was synthesized after codon redesign and optimization (Synbio Technologies, Suzhou, China). The synthesized RMgase gene was then inserted into expression vector pINA1312 to obtain the reconstructed plasmid pINA1312-RMgase. The recombinant plasmid pINA1312-RMgase was linearized with *NotI*. Linearized DNA vector containing RMgase gene was transformed into *Y. lipolytica* URA^-^ cells by the LiAc method [24]. As a control, the linearized vector pINA1312 without exogenous gene was also transformed into *Y. lipolytica* URA^-^ cells. The yeast cells were spread on the above-mentioned YNB plate to select the positive transformants and then cultured at 28 °C for 3 days. Genomics DNA from these transformants owing the relative higher α-galactosidase activity were extracted and used to amplify the RMgase gene with previous primers. Transformants were cultivated in YPD liquid medium at 30 °C for 20 h and transferred into GPPB medium for 3 days. Among a total 216 these transformants, G82 had the highest extracellular α-galactosidase activity of 121.6 U/mL.

### 2.3. Enzymatic Activity Assay and Sugar Detection

Detection of α-galactosidase activity detection was evaluated using pNPG as substrate [21]. The reaction solution was terminated by adding Na_2_CO_3_ and filtered through a 0.22-μm membrane for high-performance liquid chromatography (HPLC) analysis. An Agilent 1200 system (Agilent Technologies, Palo Alto, CA, USA) with NH_2_ column (Thermo Scientific, Sunnyvale, CA USA) was employed to detect the carbohydrates content. The carbohydrate concentration was calculated according to peak areas and retention time. The amount of enzyme releasing 1 μM *p*-nitrophenol per min at 45 °C and in buffer of pH 4.5 was defined as one unit of α-galactosidase activity (U). Furthermore, the α-galactosidase activity based on other three different substrates, including stachyose, raffinose and melibiose, were assessed. Three independent assays were carried out and the average values were calculated.

### 2.4. Effects of pH and Temperature on Purified α-Galactosidase Activity and Stability

The culture was centrifuged at 5000× *g*, and the α-galactosidase activity in the supernatant was detected (see below). The RMgase was purified by DEAE–Sepharose Fast Flow chromatography (GE Healthcare, Piscataway, NJ, USA). The effect of pH on purified enzyme activity was assessed with the enzyme equilibrated in different reaction citrate-Na_2_HPO_4_ buffer (3.0–8.0) for 2 h min at 45 °C. After the α-galactosidase enzyme solution (4 °C) was subjected to different pH values (3.0–8.0) for 2 h, the pH stability was evaluated by detecting residual activity. To further investigate the influence of temperature on activity, α-galactosidase activity detection was assayed at temperatures ranging from values (40–70 °C) with pH 4.5. Enzyme thermo stability was performed by estimating remaining activity after a 2 h incubation at pH 4.5 with different temperatures values (40–70 °C). Means and standard errors from three replicates per point are shown.

### 2.5. Sucrose Generation from Soy Molasses by α-Galactosidase Hydrolysis

The soy molasses was provided from a local soybean oil factory. The soy molasses was pretreated using the chemical method described in the previous study [16]. The soy molasses contained 14.26% (*w*/*w*) stachyose, 13.42% (*w*/*w*) sucrose, 3.32% (*w*/*w*) glucose, 1.73% (*w*/*w*) xylose, 0.78% (*w*/*w*) galactose, 2.81% (*w*/*w*) fructose. The crude fermentation broth of recombinant G82 with the activity of 121.6 U/mL, was used for soy molasses digestion. The pretreated soy molasses was hydrolyzed by α-galactosidase for 4 h at 45 °C and pH 4.5. The soy molasses hydrolysate (SMH) was collected and their composition was analyzed by HPLC. Furthermore, different dosages of α-galactosidase based on the amount of soy molasses with a range from 0 U/g to 20 U/g was added into the soy molasses to hydrolyze stachyose and raffinose completely. Sugar content was detected by HPLC analysis.

### 2.6. Optimization of SMH Supplementation at the Flask Level

The optimized two-stage bioprocess, including the first growing stage and the second producing stage, was proved to be an effective fermentation strategy for food-grade isomaltulose because of high production, elevated yield, and reduced byproduct [13]. A two-stage bioprocess together with optimization of SMH supplementation at the flask level was implemented in this research. At the first stage, the engineered strain *Y. lipolytica* S47 were inoculated in GPPB broth for 36 h. Then, SMH ranging from 400 g/L to 800 g/L were respectively added into the flask for another 48 h. The products were detected, and three replicates were used for each experiment.

### 2.7. Isomaltulose Production in 10-L Fermenter

To validate the optimized conditions obtained above, a large-scale isomaltulose production using a two-stage bioprocess was conducted in a 10-L bioreactor. Following this, 200 mL strain S47 culture was prepared and inoculated into a 10-L Biostat B fermenter (B. Braun, Melsungen, Germany) with a 6-L GPPB fermentation medium. The fermentation for isomaltulose production was conducted with agitation speed of 300 rpm, aeration rate of 5 L/min at 30 °C and pH 6.5. After the fermentation continued for 32 h, 700 g/L SMH broth was supplemented into the bioreactor to ferment another 16 h. The concentration was of the SMH broth, not of the sucrose. At 48 h, another 300 g/L SMH broth was supplemented into the bioreactor. For the analysis of SIase activity, sugar concentration, biomass and isomaltulose production during the fermentation, three parallel samples were taken from the fermenter regularly and were detected as described in a previous study [25].

### 2.8. Isomaltulose Biopurification with Ethanol-Producing Yeast Strain

To further reduce the carbohydrate byproducts and improve the isomaltulose purity, four ethanol-producing yeast strains, including *P. stipitis, S. cerevisiae, H. uvarum*, and *K. marxianu*, were used to check their ability to eliminate the residual galactose and xylose as Li description with some changes [26]. *Y. lipolytica* was used as a control. Briefly, the single colony from four yeast strains were respectively inoculated into YPD broth at 30 °C overnight as cell culture preparation. The ethanol-producing processes were carried out parallelly in 1-L fermenters. Following this, 600 mL of the final isomaltulose fermentation broth obtained in Section 2.8 was used as culture mediums for the four ethanol-producing strains in every fermenter. Then, 100 mL cell culture was transferred into the fermenter for 48 h at 28 °C under the static and anaerobic conditions. The analysis of ethanol and remaining sugars was executed by HPLC.

## 3. Results and Discussion

### 3.1. Secretory Expression of α-Galactosidase in Y. lipolytica

In this study, α-galactosidase was expected to release free sucrose from RFOs in soy molasses. Recombinant *Y. lipolytica* strain G82 had the highest extracellular α-galactosidase activity of 121.6 U/mL, which was higher than that of *Penicillium* sp. (6.03 U/mL), *Rhizopus* sp. (1.69 U/mL), *Gibberilla* sp. (1.42 U/mL), and *Absidia corymbifera* (18.7 U/mL) [27,28,29,30]. This activity obtained in this study was also much higher than that of recombinant *Escherichia coli* carrying the same α-galactosidase gene (11.03 U/mL) [29]. Figure 1a exhibited that the α-galactosidase activity using pNPG, stachyose, raffinose and melibiose as substrate was 121.6 U/mL, 46.3 U/mL, 20.7 U/mL, and 6.1 U/mL, respectively. As the recombinant α-galactosidase showed good RFOs-hydrolyzing activity, it had a potential to generate sucrose from the RFOs in soy molasses.

The effects of pH and temperature on enzyme activity and stability were evaluated according to former literature [29,31]. The assay results showed that the purified α-galactosidase had a relatively narrow optimum pH range (from 4.0 to 4.5) where more than 80% activity was preserved, and the highest enzymatic activity was observed at pH value of 4.5 (Figure 1b). The preference for acidic pH values is common for fungal α-galactosidases, which usually had an optimal range between pH 4.0 and 5.0 [32]. It was noteworthy that the α-galactosidase stability constantly improved as the pH increase in the pH range of 3.0 to 8.0; approximately 90% activity was remained when the pH reached 8.0 (Figure 1b). The Figure 1c demonstrated that the α-galactosidase activity had a continuous elevation with temperature rising and reached the maximum values at 60 °C. In addition, the results showed the α-galactosidase stability declined constantly with the increased temperature from 40 °C to 60 °C and was nearly totally inactivated at 70 °C after incubation for 2 h (Figure 1c).

Many α-galactosidases had poor pH stability. For example, the α-galactosidase from the *Talaromyces emersonii* expressed in the yeast *Pichia pastoris* lost most activity when the pH was beyond 5.5 [32]. However, the recombinant RMgase had a good pH stability, thus was more efficient for complicated bioprocess. Specially, the recombinant RMgase had a preference to acid condition and middle temperature, which could effectively prevent the appearance of nonenzymatic browning during the enzymatic catalysis process [33,34].

### 3.2. Sucrose Generation from Soy Molasses by α-Galactosidase Hydrolysis

To increase sucrose content in soy molasses, the recombinant α-galactosidase was used to hydrolyze the RFOs. A temperature of 45 °C and pH of 4.5 were selected as the appropriate conditions for the process. Under the conditions, the recombinant α-galactosidase exhibited good activity and stability. The influences of different α-galactosidase adding dosage on soy molasses composition were analyzed. Accompanied by the α-galactosidase adding dosage rising, RFOs in the soy molasses both had a remarkable reduction. The raffinose and stachyose were almost converted into sucrose and galactose when the adding dosage was beyond 15 U/g (Figure 1d). The final sucrose content in SMH was increased to 22.04% (*w*/*w*) and 1.64-fold than those in the undigested molasses.

The RFOs were hardly directly utilized by microbes, resulting in a low feedstock conversion rate [18,35]. Notably, in this study, raffinose and stachyose were converted into sucrose, rather than being simply eliminated [15,16,17,19,36,37,38]. Then the SMH can provided more sucrose and can be used as a substrate for isomaltulose production. This strategy significantly improved the value and utilization efficiency of soy molasses and reduced undesirable byproduct. Although the galactose was produced as a byproduct during the α-galactosidase hydrolysis, it would be removed in the subsequent process. Moreover, immobilization of enzyme is an effective method for achieving repeated use of enzymes and improved stability, permitting the catalysis of more substrate and reducing enzyme demand [39,40]. Developing a suitable immobilization method for this α-galactosidase would be part of our following work.

### 3.3. Isomaltulose Production from SMH in a Two-Stage Bioprocess

SMH, instead of sucrose, was supplemented into the medium for isomaltulose production in a two-stage bioprocess. Production strain *Y. lipolytica* S47 lacks invertase activity; therefore, SIase-catalyzing isomerization was the only pathway consuming sucrose in SMH. Strain S47 utilized glucose and fructose to maintain viability, which existed in SMH or was generated during SIase-catalyzing isomerization. Different SMH supplementation concentrations were investigated to estimate the optimal fermentation condition. The isomaltulose production had a continuous enhancement with the increase of SMH. The isomaltulose yield remained stable with a concentration range from 400 to 700 g/L and began to dramatically decrease at 800 g/L with a yield of 0.86 (g/g) (Table 1). Since a relatively higher production and stable yield were both achieved with the SMH supplementation of 700 g/L, this concentration was chosen for the following large-scale fermentation. The large-scale fermentation was performed in 10-L fermenter. At the first stage, SIase activity and biomass reached at 4.46 U/mL and 23.4 g/L after a 32-h cultivation (Figure 2). In the second stage, SMH of 700 g/L and 300 g/L was separately supplemented into the reactor at 32 h and 48 h. The maximum isomaltulose concentration (209.4 g/L) was achieved at 72 h, at which point the SIase activity, biomass, and yield were 6.93 U/mL, 32.9 g/L, and 0.95 (g/g), respectively (Figure 2). In the final broth, glucose and fructose were hardly detected as they were both transformed into intracellular lipids by the *Y. lipolytica* strain [13]. However, galactose and xylose could not be utilized by the *Y. lipolytica* strain, remaining a constant concentration at 17.3g/L and 96.1 g/L, respectively.

As mentioned above, sucrose in SMH was isomerized into isomaltulose during the bioprocess. The yield of 0.95 (g/g) was very close to that of 0.96 (g/g) using pure sucrose as a substrate in our previous study [13]. This yield was considerably high, while the isomaltulose yields in other studies were generally less than 0.90 (g/g) [9]. This was attributed to the good product specificity of the recombinant SIase and sucrose-not-utilizing characteristic of *Y. lipolytica* cells. Corresponding to the sucrose content in SMH, the isomaltulose production reached 209.4 g/L. This value was much higher than that of engineered *L. lactis* (36 g/L) and engineered *S. cerevisiae* (< 4 g/L) using pure sucrose as a substrate [10,12]. Cane molasses also has been used as sucrose as a substrate for isomaltulose production by engineered *B. subtilis*, with a yield of 0.924 (g/g). Due to the higher native sucrose content in cane molasses, the production of 221.6 g/L was slightly higher than that in this study [8]. However, compared with cane molasses, soy molasses had a clear advantage for its much lower price [19]. Thus, soy molasses could be used as a suitable and low-cost alternative to sucrose for efficient and low-cost isomaltulose production.

Actually, constructing a single strain with both α-galactosidase and SIase activities can simplify the processes. Undesirably, by tandemly expressing the SIase gene and galactosidase gene, the activities of both two recombinant enzymes were fairly low (data not shown). We speculated that this may be due to use of the same hp4d promoter for the expression of two genes. Thus, finding a new suitable promoter is the key point. In this study, a different strain G82 was constructed to gain high galactosidase activity.

### 3.4. Removing Sugar Byproducts Using Yeast Strains

Apart from target product isomaltulose, galactose, xylose, and trehalulose remained in the final broth, with the concentration of 96.1 g/L, 17.3g/L and 4.6 g/L, respectively. To eliminate the undesirable sugars and avoid introducing new metabolites, four ethanol-producing yeast strains were employed to perform the bioremoval process. As shown in Figure 3, galactose can be totally eliminated by *S. cerevisiae, K. marxianus and P. stipitis*; xylose can only be removed by *P. stipitis*; no loss of trehalulose and isomaltulose was observed after treatments by all the four strains. Therefore, *P. stipitis* was identified as the optimal strain for bioremoval. After a distillation process, only isomaltulose (209.4 g/L) and trehalulose (4.6 g/L) remained in the broth. On the whole, by combining enzymatic hydrolysis and fermentation, 1000 g soy molasses was converted into 209.4 g isomaltulose and 32.9 g ethanol. Based on the much higher value of isomaltulose than ethanol, the ethanol production occupied a small percentage of the total output. However, ethanol production to remove other sugars played an important role for efficient extraction of isomaltulose from the broth. As solubilities of the monosaccharides were much larger than that of isomaltulose, the residual monosaccharides of high concentrations can weaken the extraction of isomaltulose, discarding large part of the product and reducing economic value.

For functional sugar manufacturing, additional sugar byproducts were always followed with the target product. Current sugar separation methods were based on the chromatographic technique, which is not only laborious and time-consuming, but also very expensive. The use of yeast cells to remove specific sugars can be a promising alternative method. According to the discrepancy of utilizing ability for different sugars, undesirable sugar products can be consumed by the cells, while target product remained. In trehalose synthesis, the remaining maltose substrate and byproduct glucose were both converted into ethanol by *S. cerevisiae,* then a higher trehalose purity (99.5%) was obtained [24]. However, *S. cerevisiae* cannot utilize any kinds of undesirable sugars remaining in the final isomaltulose fermentation broth. Thus, non-traditional ethanol-producing yeast *P. stipitis* was screened to perform the bioremoval. After the treatments by *P. stipitis*, isomaltulose accounted for 97.8% of the total sugars, much higher than the proportions of generally below 90% in other isomaltulose productions. The results demonstrate that the bioremoval process adopted in this study can help achieve high-purity isomaltulose production. Additionally, the use of *P. stipitis* can achieve removing more kinds of specific undesirable sugars and broadened the application of ethanol-converting bioremoval method.

## 4. Conclusions

In this study, a three-step linkage strategy for isomaltulose production from soybean molasses was established. The sucrose content in the soy molasses was increased to 22.04% (*w*/*w*) by α-galactosidase hydrolysis. SMH was used as the substrate, with isomaltulose production of 209.4 g/L, and a yield of 0.95 (g/g). Finally, by virtue of the bioremoval process using *P. stipitis*, isomaltulose accounted for 97.8% of the total sugars. Therefore, this bioprocess is expected to provide a basis for further development of efficient and low-cost isomaltulose production. Construction of engineered strains performing the three steps simultaneously would simply the bioprocess.

## Figures and Tables

**Figure 1 biomolecules-09-00353-f001:**
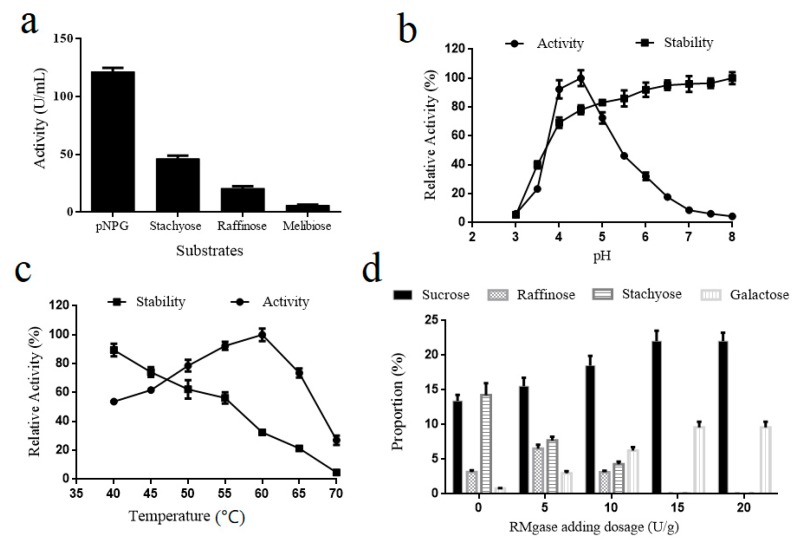
(**a**) RMgase activity based on different substrates. (**b**) Effect of different pH values on the activity and stability of the purified enzyme. (**c**) Effect of different temperatures on the activity and stability of the purified enzyme. (**d**) Effects of the amount of RMgase addition on sucrose recovery from soy molasses. Data are given as means ± standard deviation, n = 3 (b, c, d).

**Figure 2 biomolecules-09-00353-f002:**
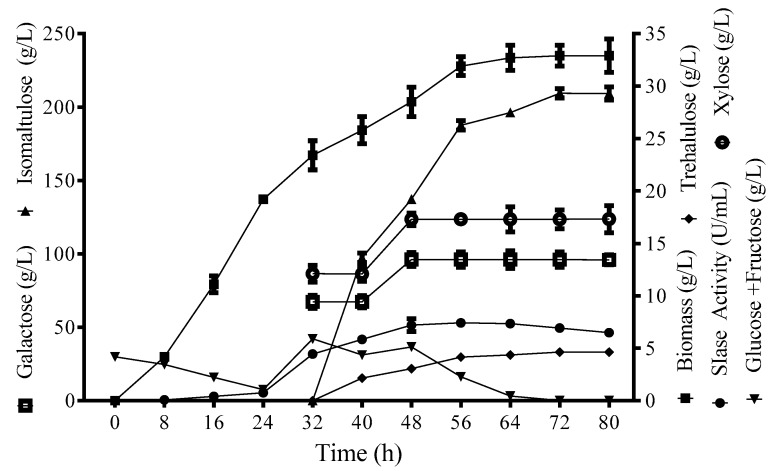
Time course of isomaltulose production, biomass, sucrose isomerase (SIase) activity, and sugar concentrations in the 10-L bioreactor during fermentation by *Y. lipolytica* S47. Data are given as means ± standard deviation, n = 3.

**Figure 3 biomolecules-09-00353-f003:**
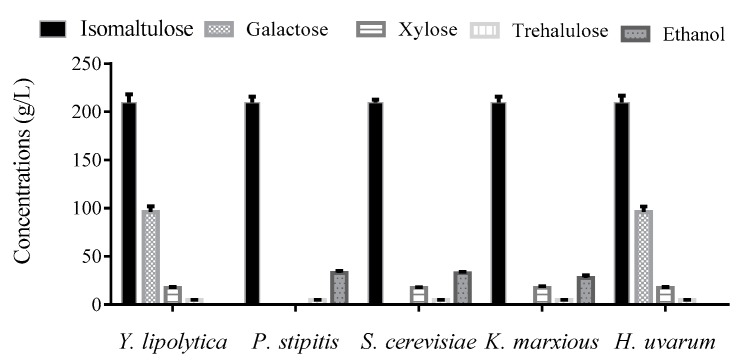
Sugar compositions and ethanol concentrations after bioremoval using different ethanol producing strains.

**Table 1 biomolecules-09-00353-t001:** Isomaltulose production from soy molasses of different concentrations.

Digested Soy Molasses (g/L)	Isomaltulose Production (g/L)	Residual Sucrose (g/L)	Yield (g/g)
400	84.7 ± 7.6	0	0.96
500	106.3 ± 8.2	0	0.96
600	126.2 ± 7.2	0	0.96
700	135.9 ± 12.1	11.6 ± 1.3	0.95
800	144.8 ± 10.9	7.9 ± 0.6	0.86

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
