# Peer review of "Whole Conversion of Soybean Molasses into Isomaltulose and Ethanol by Combining Enzymatic Hydrolysis and Successive Selective Fermentations"

_biomolecules, 2019, doi:10.3390/biom9080353_

Round 1
Reviewer 1 Report
I find interesting the paper. They show a 2 step production of isomaltulose from molasses, and the transformation of the side products to ethanol in a last step. That is, they use debris to produce an interesting product and the new production debris are used to produce ethanol. This is not properly visualized in the title. Moreover, English and style needs some improvements, the paper may be understand but not always easily.
I have just one question. Why in the first hydrolysis they prefer whole cells better that cells free and immobilized enzymes? Except if they have proper reasons, the use of immobilized enzymes (higher volumetric activity, stability, etc, add some references, reviews on immobilization) could be proposed as a way to improve these already nice results in this future. Baggase, molasses, etc have been modified using immobilized enzymes, and recently the reuse of debris have been covered in a review:
Scopus
EXPORT DATE:22 Jul 2019
Ferreira-Leitão, V.S., Cammarota, M.C., Aguieiras, E.C.G., de Sá, L.R.V., Fernandez-Lafuente, R., Freire, D.M.G.
6504613045;35570358200;25823669100;37101308800;35552449900;7003347454;
The protagonism of biocatalysis in green chemistry and its environmental benefits
(2017) Catalysts, 7 (1), art. no. 9, . Cited 19 times.
https://www.scopus.com/inward/record.uri?eid=2-s2.0-85008678038&doi=10.3390%2fcatal7010009&partnerID=40&md5=5968cc71d23016155f11775f0af93c0e
DOI: 10.3390/catal7010009
DOCUMENT TYPE: Review
PUBLICATION STAGE: Final
ACCESS TYPE: Open Access
SOURCE: Scopus
Reviewer 2 Report
I have made extensive comments in the attached file. Places highlighted red contain obvious grammatical or spelling errors. Places highlighted yellow contain my comments or questions.
I was interested in this work when skimming over the abstract initially. I still think the work has a good potential to "eventually" become industrially important but the current manuscript describes work progressing towards that finally truly optimized process for industrial usage. This itself is perfectly fine to this reviewer, IF the authors disclosed this clearly in the Introduction, Discussion, and Conclusions with clear statements of the eventual process envisioned, previous stage of development, and what exact developments pursued and achieved in this work. Unclear descriptions are not helpful.
Some of the questions (others are in the attached file):
Why producing galactosidase with a strain G82 and producing isomaltulose with a different strain S47? Are there fundamental challenges or considerations of developing one single strain and making the proposed three-stage process including enzyme production, enzymatic hydrolysis and then isomaltulose production, into a single stage process?
How was the ethanol production process integrated into the above three-stage process? I suspect it's not actually integrated in process but in concept. That's fine. But, please describe how you envision this to be done. I am curious if the authors consider this step as a byproduct (ethanol) making process or just a "separation" process to remove other sugars. Because of the large amounts of these sugars from soy molasses, thsi different consideration can strongly affect the overall economics.
Data in Figure 2 cannot be correct. I have given detailed questions in the file.
